# Exploring the Role of Ash on Pore Clogging and Hydraulic Properties of Ash-Covered Soils under Laboratory Experiments

Taehyun Kim [1], Jeman Lee [1], Ye-Eun Lee [2] and Sangjun Im [1,3,*]

[1] Department of Agriculture, Forestry and Bioresources, Seoul National University, Seoul 08826, Korea; kta40@snu.ac.kr (T.K.); jemahn@snu.ac.kr (J.L.)
[2] Division of Forest Fire and Landslide, National Institute of Forest Science, Seoul 02455, Korea; dldpdms109@korea.kr
[3] Research Institute of Agriculture and Life Sciences, Seoul National University, Seoul 08826, Korea
* Correspondence: junie@snu.ac.kr

**Abstract:** Fires can alter the hydraulic properties of burned soils through the consumption of organic matter on the ground surface. This study examined the effects of rainfall on the presence of soil pore clogging with varying ash layer thickness using laboratory rainfall simulator experiments. The image analysis with resin impregnation showed that rainfall impact caused plugging of soil pores at 22.2% with soil particles and 14.3% with ash particles on near surface soils (0–5 mm below). High rainfall intensities enhanced soil pore clogging by ash particles, particularly at shallow soil depths (0–10 mm). Ash deposits on the soil surface increased the water-absorbing capacity of ash-covered soils compared with that of bare soils. The rainfall simulation experiments also showed that ash cover led to a reduction in soil hydraulic conductivity, owing to the combined effects of surface crust formation and soil pore clogging. The complementary effects of soil pore clogging and water absorption by ash cover could hamper the accurate understanding of the soil hydrologic processes in burned soils.

**Keywords:** rainfall simulator; resin impregnation; soil hydraulic conductivity; soil pore clogging; sorptivity; surface crust

## 1. Introduction

The presence of ash on the soil surface may contribute to post-fire changes in soil hydraulic properties [1]. Combusting organic matter can intensify soil hydrophobicity near the soil surface [2] and increase the likelihood of soil pore clogging by ash particles [2–5], leading to increases in runoff and sediment yields by several orders of magnitude [6–8].

Ash refers pyroclastic deposits with mean diameters smaller than 2 mm [9], and it is produced from the combustion of surface fuels and organic matter in litter and duff layers. After a fire, ash is deposited on the soil surface, which is typically a few millimeters to several centimeters thick [10–12]. The physical and chemical properties of ash particles vary depending on plant species, combustion completeness, and fire severity [1,13], with a wide spectrum of carbon-rich black ash and carbonate-rich white ash [1].

The amount of pore space can be altered by pore clogging after a fire [5,14]. Fine ash particles can easily penetrate the soil macropores in a highly variable manner, depending on the soil texture, ash particle size, and rainfall characteristics. Post-fire pore clogging is highly likely to occur in coarse soils because large pores are vulnerable to being plugged by fine ash particles [15,16]. According to [16], ash particles easily block the pores of sandy loam but have little or no influence on small silt loam pores. Ash layers can retain a depth of water approximately equal to half their thickness for a limited time immediately after a fire [17,18], thereby enhancing the amount of water infiltrated, which offsets the reduction in infiltration capacity by pore clogging [13,16,19]. As the thickness of the ash

layer increases, the magnitude of pore clogging increases; however, the spatial extent of the area that contributes to infiltration proportionally increases [10,20].

The physical impact of raindrops is the dominant factor that blocks soil pores in ash-covered soils. Raindrops are held under positive pressure by surface tension forces and readily break apart when they strike the soil surface [21]. Raindrops that splash outward from the lying soil surface strike in the form of a fast-moving mixture, and their velocity may reach twice that of the incident [22]. This jet-like flow is responsible for breaking soil aggregates and displacing fine materials laterally and vertically. Therefore, an incident water droplet is able to transport ash particles through the soil pore spaces. Rainfall characteristics, including droplet size distribution, terminal velocity, and rain duration, can affect the presence and intensity of pore clogging in ash-covered soils.

Soil pore clogging in fire-affected soils has received relatively less attention than the presence of soil water repellency, mainly because of the practical difficulties of in situ measurements [8,13]. This is mainly because ash particles are rapidly dispersed or removed by wind and water after a fire [23,24]. The spatial variation and uncertainty in soil and ash properties obscure the influences of ash and other influencing parameters on soil hydrology. Laboratory experiments have been adopted to overcome these limitations by controlling as many factors as possible [25,26]. They quantified the effects of ash on soil hydraulic properties, overland flow, and soil erosion. However, still few studies address the presence of pore clogging on ash-covered soils and the effect of pore-blocking on soil hydraulic properties.

With the advent of advanced sensor and computer technologies, soil matrix structure or macropore topology can be visualized and quantified using non-destructive techniques, such as microscopic observations of thin sections [27] or X-ray computer-assisted tomography [28–30]. These methods have become more widely used in soil research over the last decade [31,32] and provide an accurate interpretation of soil micromorphology [33]; however, they are expensive and require specialized equipment.

Since the 1970s, soil micromorphology, coupled with image analysis, has been increasingly used in soil science [34,35]. The geometry and distribution of the soil pore architecture can be observed from images derived from thin sections or from impregnated blocks using a variety of photographic techniques [34–37]. Stoof et al. [19] used a digital microscope to visualize the movement of the ash particles. Image analysis of soil structure provides detailed insights into the physical properties and hydrological behavior of soil [34,38].

An understanding of how ash blocks the soil pore space and, subsequently, how pore clogging affects the soil hydraulic property, is a vital subject in recent studies on post-fire hydrology. The objective of this study was therefore to examine the role of ash on soil hydraulic properties under laboratory conditions. The primary research question addressed the presence and severity of pore clogging associated with ash treatment under rainfall simulation. Furthermore, the combined effects of ash cover and pore clogging on the hydraulic conductivity of the ash-covered soils were measured and analyzed.

## 2. Materials and Methods

### 2.1. Laboratory Rainfall Simulation Experiment

2.1.1. Rainfall Simulator

Rainfall simulation experiments were conducted in the laboratory to quantitatively assess the role of fire ash on soil hydraulic properties. The experimental setup comprised spray nozzles and reception soil boxes, as shown in Figure 1. Downward-facing spray nozzles were mounted on an overhead carrier of 100 cm (L) × 150 cm (W) × 350 cm (H) with a supporting metal frame. A soil reception box was placed on each quarter beneath multiple spray nozzles. The polyethylene (PE) soil box was 20 cm long, 20 cm wide, 10 cm deep, and filled with 5 cm of soil.

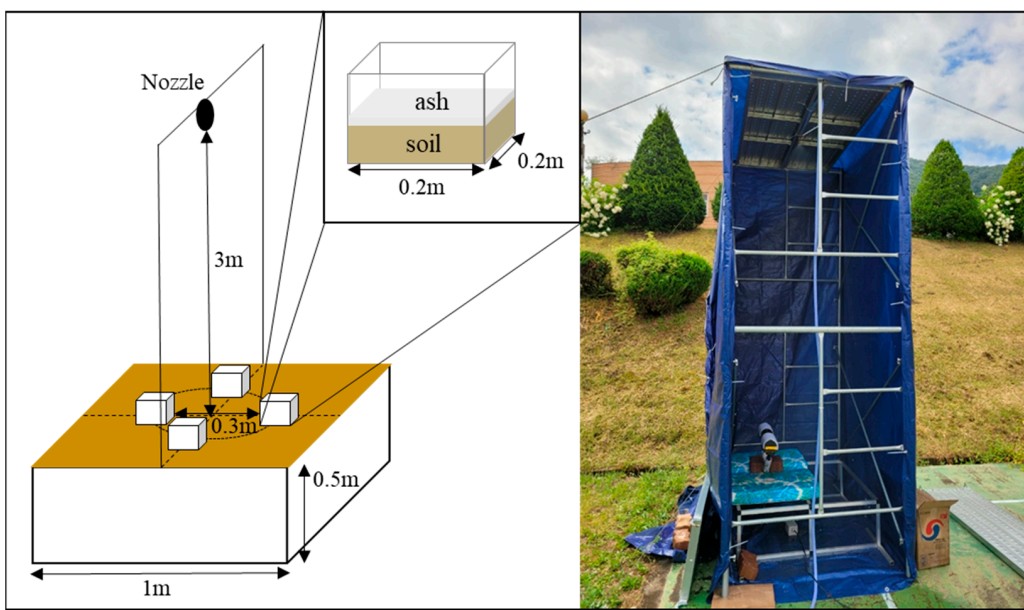

**Figure 1.** Laboratory rainfall simulation experiment.

A non-pressured dropper can provide a uniform rainfall distribution with constant fall velocity, but its use is restricted because of the difficulty in achieving a terminal velocity similar to that of natural rainfall unless the water falls from a considerable height [39]. Pressurized spray nozzles produce sufficient kinetic energy for rain droplets at a lower fall height [39,40]. Many types of spray nozzles are available for artificial rainfall simulation [40]. The full-jet nozzle (Kukje Nozzle Co., Seoul, Korea) was selected in this study, and a 12 horsepower static pressure pump (PH-450C, Hanil Electric Co., Ltd., Wonju, Korea) was used to generate a constant discharge pressure at the nozzle. The diameters and discharge of nozzles used are summarized in Table 1.

**Table 1.** Rainfall intensity and kinetic energy of simulated rainfall.

| Experiment | Return Period (Year) | Spray Nozzle | | Rainfall Intensity (mm/h) | | Kinetic Energy (J/m²/h) | Christiansen's Uniformity Coefficient |
| --- | --- | --- | --- | --- | --- | --- | --- |
| | | Diameter (mm) | Discharge (L/min) | Okgye [41] | This Study | | |
| R02 | 2 | 0.89 | 0.74 | 22.2 | $21.87 \pm 9.02$ | $35.19 \pm 15.86$ | 0.91 |
| R10 | 10 | 2.0 | 3.7 | 35.6 | $37.50 \pm 7.63$ | $83.55 \pm 20.47$ | 0.95 |
| R50 | 50 | 3.6 | 11.1 | 57.3 | $47.97 \pm 8.97$ | $134.08 \pm 30.52$ | 0.98 |

The rainfall intensity was set at a constant level for each experiment by adjusting the number of nozzles and the water pressure. Based on preliminary experiments, spray nozzles were mounted 3 m above the ground to achieve a terminal velocity similar to that of a free-falling raindrop. Raindrop distribution, terminal velocity, and the corresponding kinetic energy for each experiment were measured using a laser-based optical disdrometer (OTT Parsivel, OTT Hydromet Co., Luebek, Germany).

The characteristics of simulated rainfall are listed in Table 1. The average rainfall intensities were set at 21.87, 37.50, and 47.97 mm/h for 30 min in each experiment, which are nearly comparable to the design storms in the Okgye region with a return period of 2, 10, and 50 years, respectively (hereafter R02, R10, R50) [41]. The drop size of artificial rainfall varied between 0.1 mm and 4.0 mm, which is apparent for most moderate rainfall cases in Korea [42]. The median values of simulated raindrop were 0.8, 1.2, and 1.9 mm for R02, R10, and R50, respectively. The corresponding kinetic energies of rainfall were 35.19,

83.55, and 134.08 $J/m^2/h$, respectively. The values of the Cristiansen uniformity coefficient were 0.91 for R02, 0.95 for R10, and 0.98 for R50, which were in the acceptable range of published uncertainty [43,44].

### 2.1.2. Soil and Ash Sample Preparation

The Okgye area is in the eastern region of the Korean peninsula, and experienced a substantial forest fire in 2019, where more than 1260 ha of forest was severely damaged [45]. Soil was collected from a depth of approximately 5–10 cm at the unburned site in the Okgye area to represent the soil characteristics of fire vulnerable forests. The sampled soils were composited to ensure homogeneity and then sieved (#4 sieve) to remove coarse debris and woody particles. Six subsamples were prepared to analyze the physical and chemical properties of the soil at the National Instrumentation Center for Environmental Management (NICEM) of Seoul National University (SNU). The particle size distribution was determined using sieve analysis for particle sizes larger than 125 μm or hydrometer analysis for particles smaller than 125 μm. Organic matter content was determined using the loss-on-ignition method [46].

Table 2 lists the physical and chemical properties of the soil samples. The soil samples were predominantly sand (79.48 ± 0.41%) and were classified as loamy sand according to the USDA soil taxonomy classification system. The average porosity of soils fluctuated between 2.0% and 3.8%, which is in accordance with porosity values reported by [47].

**Table 2.** Physical and chemical properties of soil samples.

| Soil Property | Particle Size Distribution (%) | | | Soil Texture | pH | TOC (%) | Bulk Density (g/cm³) |
|---|---|---|---|---|---|---|---|
| | 2–0.05 mm | 0.05–0.002 mm | <0.002 mm | | | | |
| Soil Sample | 79.48 ± 0.41 | 16.68 ± 0.90 | 3.84 ± 0.34 | Loamy sand / SM | 5.28 ± 0.27 | 5.53 ± 1.38 | 1.25 ± 0.10 |

The fire ash sample was produced from the combustion of Korean red pine (*Pinus densiflora*) litter in a fireplace. Leaf litter of *P. densiflora*, which is a common tree species in the eastern Korean forests, including Okgye, was collected in September 2021 in the Chilbo-san University Forest of SNU and dried two weeks in the laboratory to retain a moisture content less than 20%. The litter was neither cleaned nor sorted to preserve natural field conditions.

The flame temperature was monitored using thermocouple probes during burning. The k-type thermocouple probes (K2-T-0.32, SM Instruments Co., Seoul, Korea) were placed directly above the top of the pine litter bed to measure the gas phase temperature. The ash was sieved to 2 mm to separate the fully combusted ash (<2 mm) from the partly combusted chars (2–5 mm) and then transported to the NICEM for analysis. The size distribution of the ash particles was determined using a laser diffraction analyzer (Malvern Instruments Ltd., Malvern, UK), and the total carbon content of the ash was analyzed by the loss-on-ignition method [46]. The ash pH was measured using a Crisol GLP 22 pH meter [48].

The ash particles were found to have a high alkalinity with a pH of 10.7 ± 0.21 and total organic carbon content of 4.58 ± 1.02%. Ash color is one of the most visible and diagnostic features of combustion completeness, which may be related to temperature, duration of burning, and $O_2$ availability [13]. The ash color was grayish, indicating a high-severity combustion. The mean temperature of the flame was approximately 530 °C, reaching a maximum of 890 °C.

The size distribution of the soil and ash particles were likely major contributors to pore clogging [49], implying that coarse ash particles do not easily penetrate into small soil pores. Figure 2 showed that the mean particle size of the ash (D50) was 0.5 mm, which lies within the typical range of the ash particles reported [50,51]. The particle size of the pine needle ash was likely dependent on the combustion temperature, with finer white ash being correlated with more complete combustion [1,16].

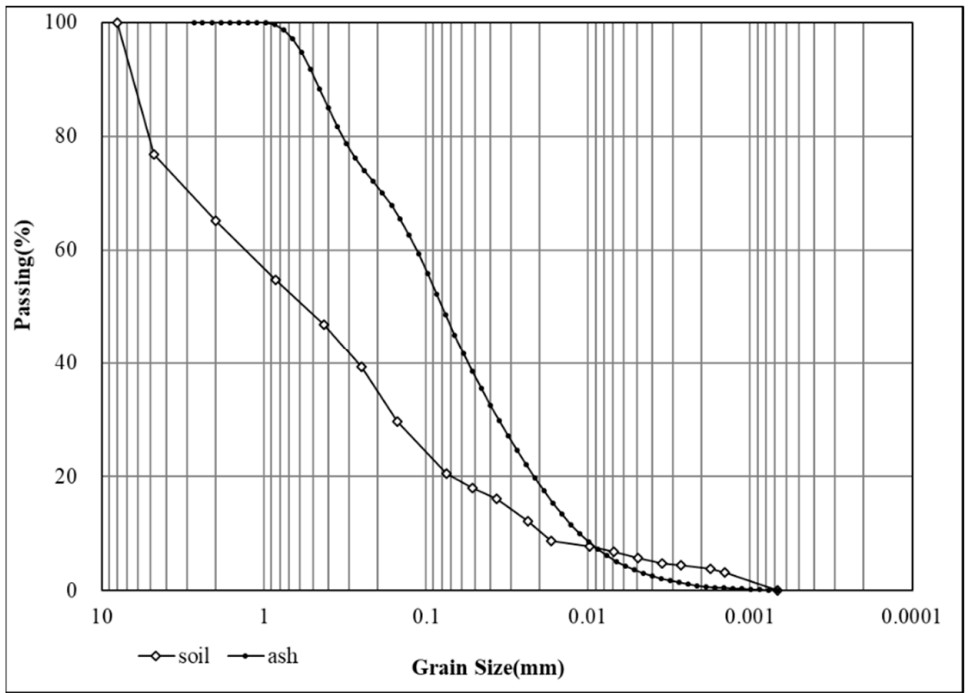

**Figure 2.** Particle distribution of soil and ash.

2.1.3. Experiment Design

Given that forest fires occur in the dry season in South Korea, air-dried forest soil was used in the experiments, which has a moisture content of 20% and a particle density of 2.45 g/cm$^3$. The reception soil box was manually filled with air-dried soil and then placed in the laboratory for 48 h to achieve a bulk density of 1.25 g/cm$^3$, which is a target density representative of the unburned soil in Okgye, South Korea (Table 2).

Field measurements in the burned forest of South Korea showed that the mean ash depth was 15 $\pm$ 0.56 mm (n = 52), which is within the typical range recorded in previous studies [18,20]. Therefore, three ash treatments were undertaken in this study, namely bare soil with no ash cover, a low-ash treatment of 5 mm thickness, and a high-ash treatment of 20 mm thickness. Rainwater was sprayed on the ash-covered soils at three rainfall intensities (2-, 10-, and 50-year return period) for a 30 min duration (Table 1). A no-rainfall experiment was conducted as the control. The laboratory experiments included three ash treatments for each rainfall condition, with six replications of each treatment for a total of 72 experiments (4 rainfall intensities $\times$ 3 ash treatments $\times$ 6 replications). All the experiments were performed in a climate-controlled laboratory with air temperatures ranging between 16 °C and 17 °C and with a relative humidity between 30% and 40%.

*2.2. Minidisk Infiltrometer Measurement*

A minidisk infiltrometer (Decagon Devices 2012) test was conducted to determine the soil sorptivity and hydraulic conductivity of the ash-covered soils after completion of the rainfall simulation experiments. This test is widely used for measuring soil hydraulic properties because of its low price, small dimensions, and ease of use [52–54]. The measurements were performed according to the technical manual (Decagon Device 2012) and previous studies [54,55]. The volume of water infiltrated was recorded at a time interval of 30 s for no less than 5 min under a suction head set at 3 cm.

The hydraulic properties were estimated using the method proposed by [55]. The cumulative infiltration values were fitted to a two-term equation with time, using Equation (1).

$$I = C_1 \sqrt{t} + C_2 t \tag{1}$$

where $I$ is the cumulative infiltration (cm), and $t$ is the time (s). $C_1$ and $C_2$ are the hydraulic parameters related to soil absorption capacity (cm/s$^{1/2}$) and hydraulic conductivity (cm/s), respectively.

The hydraulic conductivity ($k$) of the soil at a particular tension was calculated using the following Equation (2).

$$k = \frac{C_2}{A} \tag{2}$$

where $C_2$ is the slope of the curve of the cumulative infiltration versus the square root of time, and $A$ is the value corresponding to the van Genuchten parameters for a given soil type to the suction rate and radius of the infiltrometer disk. $A$ is computed from

$$A = \frac{11.65\left(n^{0.1} - 1\right)\exp[2.92(n - 1.9)\alpha h_0]}{(\alpha r_0)^{0.91}} \quad n \geq 1.9 \tag{3}$$

$$A = \frac{11.65\left(n^{0.1} - 1\right)\exp[7.5(n - 1.9)\alpha h_0]}{(\alpha r_0)^{0.91}} \quad n < 1.9$$

where $n$ and $\alpha$ are the van Genuchten parameters of the soil [56], $r_0$ is the disk radius, and $h_0$ is the suction at the disk surface.

### 2.3. Pore Clogging Measurement

#### 2.3.1. Resin Impregnation

The effects of raindrops on soil pore clogging were quantified using a resin impregnation technique. Intact soil cores were gently extracted from the soil reception box to a depth of 5.1 cm and a diameter of 5 cm when the rainfall experiments were completed. Samples were collected carefully to ensure minimal soil disturbance.

Impregnation was undertaken by sequentially replacing the pore space with the resin compound. Prior to resin impregnation, the soil samples were dehydrated in the oven at 70 °C for 48 h [36]. The resin was poured into the dried soil samples until it filled the empty soil pores. Epoxy resin is particularly suitable for hardening soils because the curing time is substantially less than that for polyester resin and it does not need to be kept in an oven to produce solid samples [57]. An epoxy resin mixture was prepared by mixing 300 g of resin, 90 g hardener (EpoxAcast$^{\text{TM}}$ 690, Smooth-On Inc., Macungie, PA, USA), and blue dye (Youngnam Corp., Bucheon, Korea). The soil castings were left at room temperature for 48 h to solidify. After hardening, the soil samples were cut at the appropriate depth below the soil surface (0, 5, 10, 15, and 20 mm below) and gently ground using a mechanical saw until the surface of the soil block was smooth with as little relief as possible [58].

#### 2.3.2. Pore Image Analysis

Image analysis is a fast and effective method to measure the presence of soil pores in soil science [59]. A digital microscope (Dino-Lite Digital Microscope, Namyangju, Korea) was used in this study to acquire 0.5 cm × 0.8 cm images at depth intervals of 5 mm in the resin-impregnated soils. The images comprised a 1920 × 1024 matrix of picture elements (pixel) and were taken from five horizontal sections at each soil depth. Each pixel has a distinct color where the soil particles are white, the ash particles are black, and the pores have blue color features (Figure 3).

The extent of the soil pores, expressed as volume density, was digitally quantified by distinguishing between the strained and unstrained pixels [60,61]. Volume density represents the fraction of pores that occupy a certain area of the soil matrix [34]. A high value for volume density indicates a large portion of soil pore where pore plugging is not well developed.

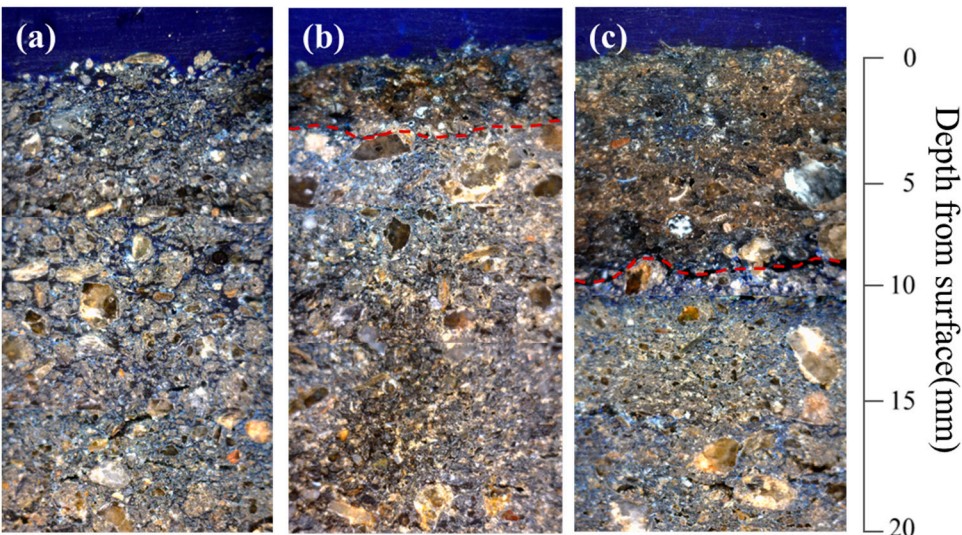

**Figure 3.** Images for resin-impregnated soil section with 37.50 mm/h rainfall(R10) for (**a**) bare soil (D00), (**b**) 5 mm ash-covered soil (D05), and (**c**) 20 mm ash-covered soil (D20).

*2.4. Data Analysis*

The rainfall experiment was replicated six times for each treatment, and the image analysis for measuring the soil pore clogging was repeated at five different locations for each given depth. We used ANOVA and Tukey's post-hoc test to compare the degree of soil clogging with varying rainfall intensity and ash thickness at a significance level of $\alpha = 0.01$. All statistical analyses were performed using the R-package, version 4.1.2 (2021-11-01) [62].

## 3. Results

### 3.1. Presence and Severity of Soil Pore Clogging

Pore plugging by ash particles was visualized with the aid of image analysis (Figure 3). Given that ash particles could not be distinguished from the soil particles based on color, the extent of ash clogging was indirectly estimated based on differences in the percentage of stained (blue) pixels between the experiment. The volume densities for the bare soils (no ash treatment) were considered as the references of soil macropores for each rainfall simulation.

Figure 4 shows the volume density of the impregnated soil sections for soil depth. Volume densities of bare soil samples are also presented for comparison. The volume density of bare soils (n = 6) under no rainfall decreased substantially with soil depth, from $56.1 \pm 1.3\%$ at a depth of 0–5 m to $24.4 \pm 4.0\%$ at 15–20 mm, mainly due to natural consolidation.

The differences in the volume density of the soil pores between the no rainfall and rainfall experiments demonstrated the effects of the raindrop impact on the migration of fine particles through the soil macropores. The occurrence of pore clogging in the bare soils was observed during the rainfall experiments. Raindrop energy breaks the soil aggregates and pushes disaggregated particles through the soil macropores [63,64]. The volume densities for the bare soils were 34.3% for R02, 33.7% for R10, and 33.6% for R50 at 0–5 mm soil depth. Rainfall had a distinct effect on pore clogging by soil particles, particularly near the soil surface. The two-fold increase in rainfall intensity (from R50 to R02) caused an approximate 0.7–4.1% additional reduction in volume density with soil depth because of pore clogging by soil particles. With the increase in rainfall intensity for the bare soils, the volume density of the soil pores gradually decreased; however, the effects of rainfall intensity on pore plugging was not significant.

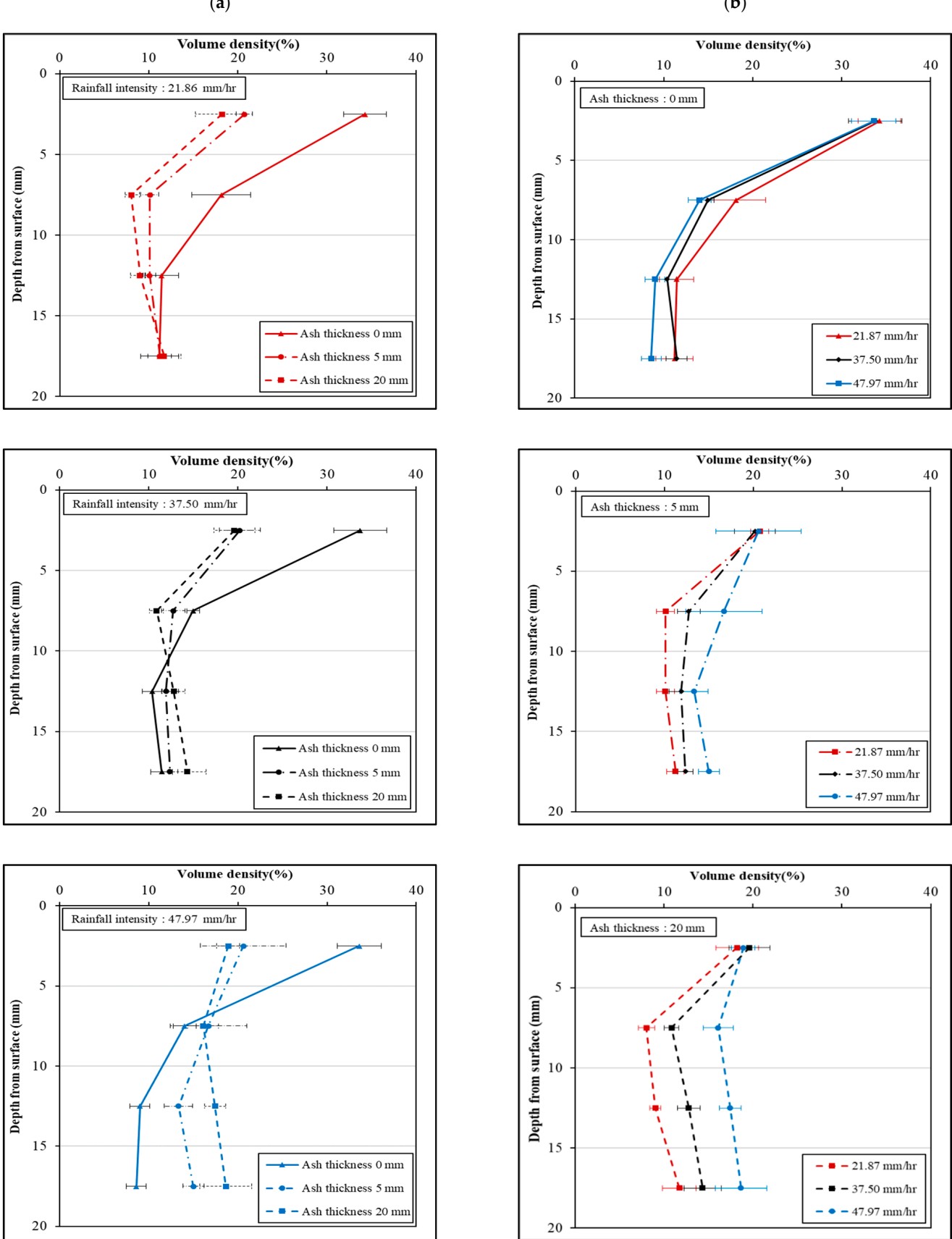

**Figure 4.** Variations in volume density according to (**a**) ash treatment and (**b**) rainfall intensity.

By comparing the extent of soil pores between the bare soil and ash treatments (Table 3), additional reduction in pore volume because of ash clogging was observed. The volume density in the upper soil layer (0–5 mm) decreased from 34.3% to 20.7% for the low-ash treatment and 18.2% for the high-ash treatment under the R02 experiment. A similar trend was observed for the R10 and R50 rainfall experiments. The additional reduction in pore volume was contributed to pore clogging by the ash particles. The ash deposition on the soil surface caused a 13.6% and 14.7% pore reduction for the low- and high-ash treatments, respectively, at a soil depth of 0–10 mm, irrespective of the rainfall intensity.

**Table 3.** Volume density of soil pore with ash treatment and rainfall intensity.

| VolumeDensity (%) | | R0 | R2 | | | R10 | | | R50 | | |
|---|---|---|---|---|---|---|---|---|---|---|---|
| | | D00 | D00 | D05 | D20 | D00 | D05 | D20 | D00 | D05 | D20 |
| Soil Depth (mm) | 0–5 | 56.1 ± 1.3 | 34.3 ± 2.4 | 20.7 ± 0.9 | 18.2 ± 2.4 | 33.7 ± 3.0 | 20.2 ± 2.3 | 19.1 ± 2.3 | 33.6 ± 2.5 | 20.6 ± 4.8 | 18.9 ± 1.3 |
| | 5–10 | 38.1 ± 3.1 | 18.1 ± 3.3 | 10.1 ± 1.0 | 8.0 ± 0.9 | 15.0 ± 0.7 | 12.8 ± 1.3 | 10.5 ± 0.8 | 14.0 ± 1.3 | 16.7 ± 4.3 | 16.1 ± 1.7 |
| | 10–15 | 27.7 ± 4.2 | 11.4 ± 1.9 | 10.1 ±0.7 | 9.0 ± 0.6 | 10.4 ± 1.1 | 11.9 ± 1.4 | 12.3 ± 1.3 | 9.0 ± 1.3 | 13.3 ± 1.6 | 17.4 ± 1.2 |
| | 15–20 | 24.4 ± 4.0 | 11.2 ±2.1 | 11.2 ±1.3 | 11.7 ± 1.9 | 11.4 ± 1.2 | 12.4 ± 0.9 | 14.1 ± 2.1 | 8.6 ± 1.1 | 15.0 ± 1.2 | 18.6 ± 2.9 |

A slight reduction in pore volume was exhibited at the 5–10 mm soil layer for the R02 and R10 experiments, but the proportion of soil pores increased in the same soil layer for the R50 experiment. The magnitude of ash clogging intensified with the increasing addition of ash; however, this trend did not always occur. As shown in Figure 4, ash thickness had negative impacts on the formation of pore clogging at higher soil depth under high rainfall intensity.

This implies that ash overlays the exposed mineral soils and diminishes the striking force of the rain drops reaching the soil surface by forming an ash blanket or mat. Rainfall greater than the infiltration capacity of the soil–ash interface ponded on the soil surface. These combined effects can resist the disaggregation of soils and hampered the movement of fine-grained particles through the soil layer [65–68]. The bare soil treatment had no sign of ponding and did not exhibit an increase in volume density, even with high rainfall events.

### 3.2. Hydraulic Conductivity and Sorptivity

The hydraulic properties of ash-covered soils, such as hydraulic conductivity and sorptivity, were determined based on MDI measurements for a pressure head of −3 cm. Table 4 presents the hydraulic conductivities and sorptivities derived from the cumulative infiltration curves across varying timespans. Coefficient k in Equation (2) and $C_1$ in Equation (1) refer to the estimated values for hydraulic conductivity and sorptivity, respectively.

The MDI measurements with the no rainfall condition reflected the blanket effect of ash on soil hydrology. The average values of hydraulic conductivity and sorptivity for the bare soil under no rainfall conditions were $0.0104 \pm 00013$ (n = 6) cm/s and $0.0409 \pm 0.0052$ (n = 6) cm/s$^{1/2}$, respectively. The bare soil generated substantially greater hydraulic conductivity and sorptivity than the ash-treated soil under the no rainfall simulation, likely because ash is hydrophobic, thereby inhibiting the penetration of water through the ash layer.

The effect of pore clogging and the ash blanket on soil hydrology was assessed by comparing the hydraulic conductivity and sorptivity of bare soils under rainfall simulation with those of the control (no rainfall) experiment. Rainfall led to changes in the hydrologic properties of the soil because of the physical sealing by the soil particles. The hydraulic conductivity and sorptivity of the bare soil decreased by 0.0036 cm/s and 0.0142 cm/s$^{1/2}$

for R02, 0.0042 cm/s and 0.0163 cm/s$^{1/2}$ for R10, and 0.0036 cm/s and 0.0139 cm/s$^{1/2}$ for R50, respectively. Tukey's test showed that the changes in the hydraulic conductivity and sorptivity of the bare soils with different rainfall intensities were not significant ($\alpha$ = 0.01).

**Table 4.** Hydraulic conductivity and sorptivity of ash-covered soils.

| Experiment | | Hydraulic Conductivity (cm/s) | Reduction Rate (%) | Tukey's HSD | Sorptivity Coefficient (m/s$^{1/2}$) | Reduction Rate (%) | Tukey's HSD |
|---|---|---|---|---|---|---|---|
| R00 | D00 | 0.0104 ± 0.0013 | | c | 0.0409 ± 0.0052 | | b |
| | D05 | 0.0059 ± 0.00089 | | ab | 0.0231 ± 0.0035 | | ab |
| | D20 | 0.0082 ± 0.0021 | | bc | 0.0324 ± 0.0083 | | ab |
| R02 | D00 | 0.0036 ± 0.0004 | −65.1 | a | 0.0142 ± 0.0014 | −65.3 | a |
| | D05 | 0.0023 ± 0.0004 | −61.5 | a | 0.0194 ± 0.0035 | −16.0 | a |
| | D20 | 0.0025 ± 0.0002 | −70.3 | a | 0.0212 ± 0.0018 | −34.6 | a |
| R10 | D00 | 0.0042 ± 0.0006 | −59.8 | a | 0.0163 ± 0.0025 | −60.1 | a |
| | D05 | 0.0019 ± 0.0002 | −67.8 | a | 0.0165 ± 0.0016 | −28.6 | a |
| | D20 | 0.0029 ± 0.0006 | −65.4 | a | 0.0248 ± 0.0051 | −23.5 | ab |
| R50 | D00 | 0.0036 ± 0.0003 | −65.9 | a | 0.0139 ± 0.0014 | −66.0 | a |
| | D05 | 0.0030 ± 0.0004 | −48.7 | a | 0.0262 ± 0.0038 | +13.4 | ab |
| | D20 | 0.0034 ± 0.0003 | −59.2 | a | 0.0293 ± 0.0027 | −9.6 | ab |

For the Tukey's HSD column, values not sharing a letter are significantly different ($\alpha$ = 0.01).

In line with the results of soil pore clogging, ash treatments had a pronounced effect on hydraulic conductivity. The hydraulic conductivity measured after the R02 rainfall simulation varied from 0.0036 (± 0.0003) cm/s for the bare soils to 0.0023 ± 0.0004 cm/s for the low-ash treatments, and 0.0025 ± 0.0002 cm/s for the high-ash treatment soils. A comparison between the ash treatments showed that the hydraulic conductivity of the ash-covered soils slightly increased as the thickness of the ash increased; however, the differences were not significant (Figure 5). Rainfall impact reduced the hydraulic conductivity, but the differences among the rainfall intensities were not significant according to Tukey's test at a significance level of $\alpha$ = 0.01. A decrease in hydraulic conductivity with increasing rainfall intensity is in line with the findings of other similar studies [1,69].

The sorptivity of the ash-covered soil is proportionally linked to the ash thickness, because thick ash layers are more capable of absorbing rainfall (Figure 6). Our results are in line with previous findings on this [16,23]. Table 4 shows that the mean sorptivity for all the rainfall simulations increased from 0.0207 m/s$^{1/2}$ for the low-ash treatment to 0.0251 m/s$^{1/2}$ for the high-ash treatment. High rainfall intensity further enhanced the water storage capacity of the ash layer. The differences in sorptivity between the bare soils and the ash-covered soils were significant for the high rainfall simulation (R50); however, there was no significant difference in sorptivity between treatments for the low-rainfall R02 and R10 simulations.

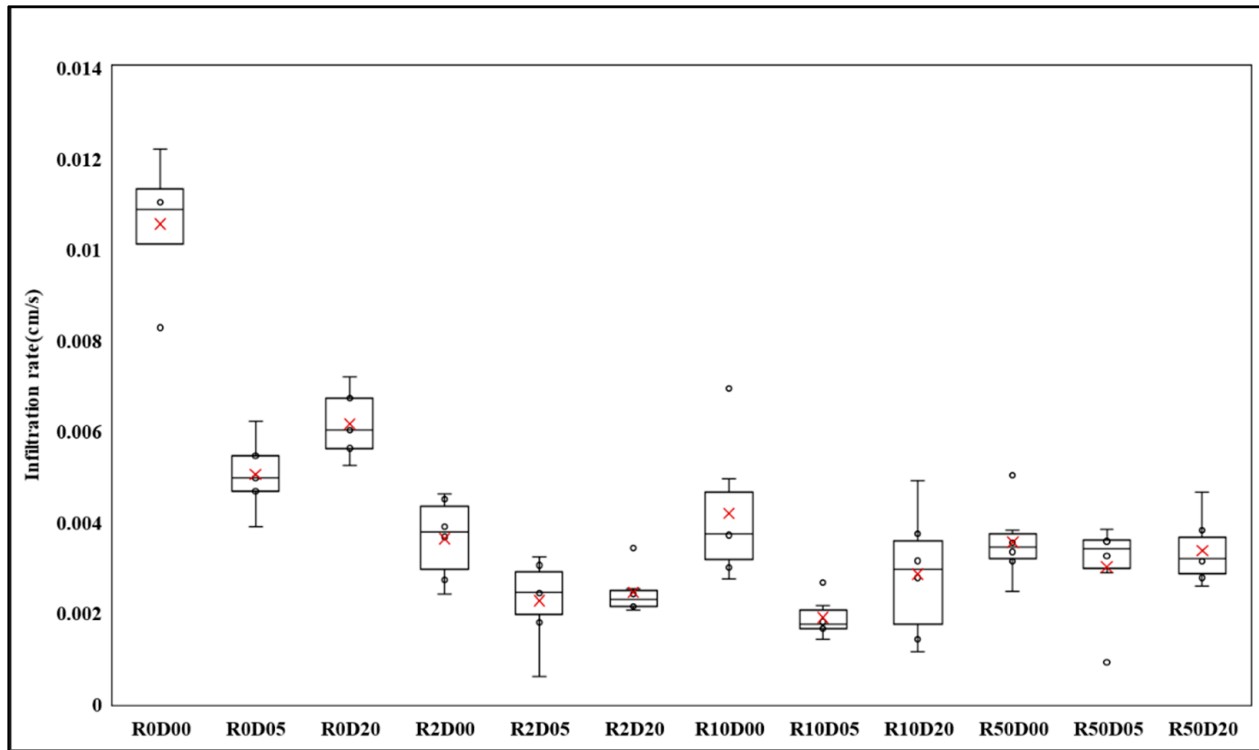

**Figure 5.** Variation of hydraulic conductivity with ash treatment and rainfall intensity.

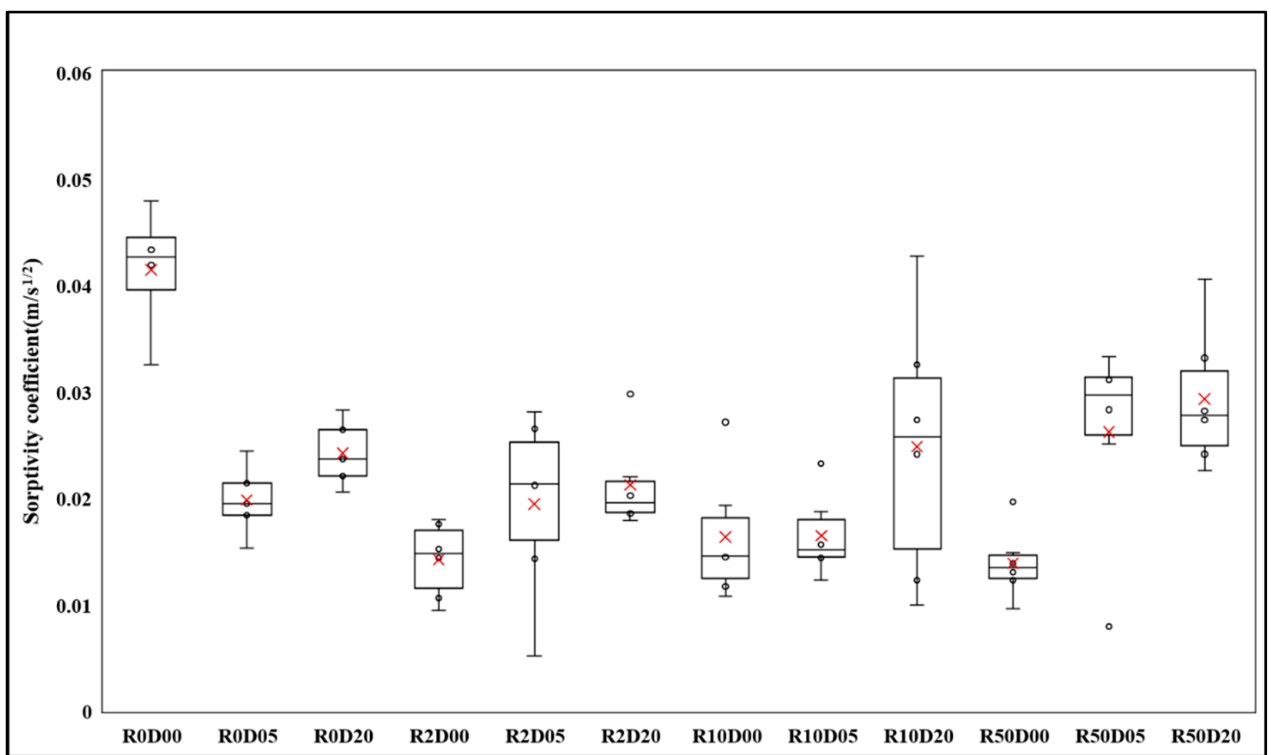

**Figure 6.** Variation of sorptivity with ash treatment and rainfall intensity.

## 4. Discussion

### 4.1. The Use of the Resin Impregnation Technique

The presence of ash particles in the soil macropores was analyzed with the aid of soil micromorphology. Soil micromorphology involves the examination of thin sections of soils

under a polarizing microscope and is a promising method for characterizing and identifying fine materials buried in soil microstructures. The main difficultly of soil micromorphology is the preparation of undisturbed soil samples [58]. There are several methods to overcome this problem. Rectangular tin or cylindrical core samples can be obtained by carefully drilling into the required parts of the soil [34,70]. This method is simple but is not suitable for use in coarse-textured soils where the structure is too loose to be collected without extensive disturbance [34]. An alternative technique is the resin impregnation method, which has been widely used to obtain high-quality images of loose and friable soils [71,72].

Resin impregnation is more expensive and labor-intensive than some traditional methods of measuring soil structure but offers the advantage of a more detailed assessment of soil pore distribution when incorporating image analysis [60]. In this study, the soil pore space was graphically identified using binary gradients between the stained and unstained pixels.

The appropriate way to quantify the soil pore clogging is to directly identify ash particles that were stuck in soil macropores using the scanning electron microscope and X-ray-computed tomography techniques. Due to the high cost for operation and a wide variation of pore geometry in soil samples, it is not always possible to accurately measure the presence and magnitude of pore blocking. Instead, indirect measurement with image analysis is often employed for the appearance analysis of pore space.

This paper proposed the reliable procedure to extract the pore space from digital camera images, assuming that the extent of soil pore is roughly the same for all soil section specimens. However, there were several limitations for the quantification of pore clogging. Ash particles sometimes cannot be distinguished from soil solids in resin-impregnated images. This led to small errors in the estimation of pore space because the epoxy resin permeated near the edge of soil particles. In addition, it should be noted that the inter- and intra-particle pore spaces have not been examined, which contribute to the hydraulic properties of soils. A further study is needed to improve the method of soil digital image acquisition and color information extraction.

The presence of pore clogging is related to the size distribution of the soil pores and fine particles. Stoof et al. [19] found that silt loam soils with large pores were not plugged by fine soil particles because the particles were too small to plug the porous conduit in the soils. This was not verified in our experiments because the size of the individual soil pore was not measured. However, the blocking of soil pores, to a great extent, was observed in the upper soil layer (0–10 mm depth), irrespective of rainfall intensity and ash treatment. Soils covered with a thin layer of ash underwent soil clogging more easily than soils with thick ash cover due to the high impact of raindrops reaching the soil surface [65–68]. In this study, the extent of soil clogging near the soil surface (0–10 mm depth) increased with increasing ash thickness; however, in deep soils (10–20 mm depth), soil clogging decreased as ash cover thickness increased. Surface ponding from excess water was observed with a thicker ash layer, reducing the kinetic energy of falling raindrops [1,19].

The surface crusting of ash is known to influence soil hydrology. However, its effects on soil hydrology are poorly understood [73]. Visual examination revealed the formation of an ash crust over the surface after rainfall simulation. The physical crusts comprised a thin layer of tightly packed clay and silt-sized ash particles, overlain by loose coarse soil. The surface crust reduced the porosity and hydraulic conductivity of the ash-covered soils. As with the pore clogging process, the low hydraulic conductivity of the ash crust influenced the infiltration capacity of soils immediately after measurement. Based on the slaking index obtained from the aggregate stability in water method [74], the ash crust was highly stable with water; however, the relationship between the crust stability and the hydraulic conductivity of ash-covered soils was not fully verified in this study.

### 4.2. Influence of Pore Clogging on Soil Hydrology

Across all the rainfall intensities, there was a positive correlation between rainfall intensity and ash clogging near the soil surface (<10 mm depth). However, the results

were less strongly correlated for the deeper soil layers (>15 mm depth) because the rainfall kinetic energy did not reach this area. The water-storing capacity of the ash layer increased with ash thickness, and the water storage likely prevented and reduced runoff [10].

The effects of pore clogging on soil hydraulic properties were clarified by rainfall simulation experiments under fully controlled environmental conditions. However, the dimensions of the experimental plots were limited, and the simulated rainfall was too short in duration to completely saturate the soil macropores. Another limitation of such experiments is that they do not reproduce the vertical translocation of ash particles along the soil pore matrix caused by preferential flow. Image analysis for identifying soil pores, incorporating resin impregnation, was used in this study. Soil pores were quantitatively assessed by distinguishing between the stained and unstained features. The image analysis showed that soil pores were plugged by moving particles of soil and ash materials, but the location and particle type resulting in pore clogging were not investigated in this study. Advanced techniques such as dye trace methods are needed for further analysis to separate the dye-stained ash particles from the soil aggregates.

Soil infiltration in no-ash-covered soils decreased as the rainfall intensity increased. This may be attributed to the formation of a surface crust. The formation of a physical crust on a soil surface can reduce the hydraulic conductivity of the soil surface and decrease infiltration into the soil. The soil crust is well developed under high rainfall intensities [75,76].

The influence of pore clogging on soil hydrology can be validated by comparing the capability of the soil to infiltrate and store water through the ash–soil matrix, which is regarded as a two-layer interface. It has been highlighted that soil clogging and the surface blankets formed by ash play complementary roles in soil hydrology. It is difficult to fully comprehend the hydrological processes of ash-covered soils after a fire. Understanding the critical role of soil sealing in burned areas is of vital importance for implementing post-fire rehabilitation treatments.

## 5. Conclusions

We conducted a series of laboratory rainfall simulation experiments to quantify the presence of soil pore clogging and its effects on the hydraulic properties of ash-covered soils. Rainfall simulations with three ash treatments (no ash, low-ash treatment of 5 mm, and high-ash treatment of 20 mm) showed that rainfall induced structural surface sealing by plugging soil pores and reduced the hydraulic conductivity of ash-covered soils. However, the effect of ash thickness on soil hydraulic conductivity was not significant irrespective of the rainfall intensity. The water-absorbing capacity of ash cover soils increased as ash thickness increased; however, the differences among different rainfall intensities and ash treatments were not statistically significant.

Detailed knowledge of the extent, size, and structure of soil pores provides important data on the physical and hydraulic properties of fire-affected soils. Digital image methods can allow for a sophisticated analysis of the soil pore structure. The direct measurement of ash transport and deposition mechanisms remains beyond the capability of the image analysis used because a clear distinction between the soil and ash particles from resin-impregnated sections is not possible.

This study provides unique experimental insights into the formation of pore clogging and its influence on soil hydrology in burned areas. These results have vital implications for the recovery of fire-affected forests and for developing effective post-fire rehabilitation techniques.

**Author Contributions:** Conceptualization, T.K. and S.I.; methodology, T.K. and S.I; software, J.L.; validation, T.K. and J.L.; formal analysis, T.K. and S.I.; investigation, Y.-E.L. and S.I.; resources, Y.-E.L.; data curation, T.K. and J.L.; writing—original draft preparation, T.K.; writing—review and editing, S.I.; visualization, J.L.; supervision, S.I.; project administration, S.I.; funding acquisition, S.I. All authors have read and agreed to the published version of the manuscript.

**Funding:** This work was supported by the National Research Foundation of Korea (NRF) grant funded by the Korea government (MSIT; NRF-2019R1A2C1089203).

**Informed Consent Statement:** Not applicable.

**Data Availability Statement:** Not applicable.

**Acknowledgments:** We acknowledge the National Instrumentation Center for Environmental Management (NICEM) of Seoul National University, Korea for the analysis of soil and ash properties.

**Conflicts of Interest:** The authors declare no conflict of interest.

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
