# Peer review of "Exploring the Role of Ash on Pore Clogging and Hydraulic Properties of Ash-Covered Soils under Laboratory Experiments"

_fire, doi:10.3390/fire5040099_

Round 1
Reviewer 1 Report
Manuscript ID: fire-1787737
Title: Presence of Soil Pore Clogging and Its Effect on Post-Fire Soil Hydrology under Laboratory Experiments
In this study, the authors have performed a study on the effects of rainfall on post-fire soil hydrology with varying ash layer thickness using laboratory rainfall simulator experiments. The objective of the study is worth investigating and the results presented are interesting. The authors have given a good background of the study and introduced their objectives nicely. The manuscript meets the standard of the journal and I recommend it for acceptance. I have a few comments below:
1. Please provide a better description of the soil reception box and the metal soil box using pictures for easier understanding. Currently, it is difficult to visualize their locations.
2. “The flame temperature at the top of the pine litter bed was monitored using thermo-couple probes during burning.” How were thermocouples placed (inserted into the soil at the top?) and were they measuring the solid-phase or gas-phase temperatures? Also provide the description of thermocouples (bead size, type).
3. What was the discharge pressure at the nozzle? Please provide calculations how does it relate/correspond to the velocity of actual rainfall drops.
4. Please specify better picture for Figure 3 and show comparison with bare soil and different rain speeds. It's again difficult to visualize. Also, clarify that it is a side cross-section view and some kind of markup (down arrow) on the side showing it down the soil to avoid possible confusion with the top view.
5. Figures can be improved. Current symbol size and legend are hard to distinguish.
6. “The ash was sieved to 2 mm to separate the fully com-busted ash (< 2 mm) from the partly combusted chars (2–5 mm), and then transported to the NICEM for analysis.” How authors came to this conclusion of selecting 2 mm value and why this was done. This will disturb the natural conditions. Please comment.
7. A suggestion to remove unnecessary/similar references and include only which have been actually used for this work. Current reference number (93) is very high for a journal article.
Author Response
Comments from Reviewer #1
1. Please provide a better description of the soil reception box and the metal soil box using pictures for easier understanding. Currently, it is difficult to visualize their locations.
[Response] Thanks for your suggestion. We redrew figure 1, including soil reception boxes. We found a mistake for the description of soil reception box, and revised it. It is made of polyethylene, not metal (See Figure 1)
2. “The flame temperature at the top of the pine litter bed was monitored using thermo-couple probes during burning.” How were thermocouples placed (inserted into the soil at the top?) and were they measuring the solid-phase or gas-phase temperatures? Also provide the description of thermocouples (bead size, type).
[Response] Thanks for your comments. We rephrased the sentences in order to clearly describe the temperature measurement equipment. “The flame temperature was monitored using thermocouple probes during burning. The k-type thermocouple probe (K2-T-0.32, SMInstrument Co., Korea) were placed directly above the top of the pine litter bed to measure the gas phase temperature.” And we provided the information on thermocouple, including the company, country, and model-type.
3. What was the discharge pressure at the nozzle? Please provide calculations how does it
[Response] Thanks for your comments. We used different types (diameter, nozzle flow, etc) of nozzle for each experiment according to the desired rainfall intensity. The diameters of nozzles were 0.89 mm (R02), 2.0 mm (R10), and 3.6 mm (R50). The corresponding nozzle discharge was also varied form 0.74 L/min for R02, 3.7 L/mm for 10 to 11.1 L/min for R50. We put the information on nozzles in Table 1, and then revised the sentence as like “The diameters and discharge of nozzles used are summarized in Table 1.”
4. Please specify better picture for Figure 3 and show comparison with bare soil and different rain speeds. It's again difficult to visualize. Also, clarify that it is a side cross-section view and some kind of markup (down arrow) on the side showing it down the soil to avoid possible confusion with the top view.
[Response] Thanks for your comment. We redrew Figure 3 in order to
5. Figures can be improved. Current symbol size and legend are hard to distinguish.
[Response] Thanks. We redrew the figures based on your comments. Please see Figures in the text
6. “The ash was sieved to 2 mm to separate the fully combusted ash (< 2 mm) from the partly combusted chars (2–5 mm), and then transported to the NICEM for analysis.” How authors came to this conclusion of selecting 2 mm value and why this was done. This will disturb the natural conditions. Please comment.
[Response] The term ‘ash’ is generally used for pyroclasts with mean diameters smaller than 2mm (Schmidt 1981). And, Stoof et al. (2016), Woods and Balfour(2010), León, et al., 2013 examined that the particle size of ash influenced on the presence of pore clogging in soils. Thus, we used the completely combusted ash with less than 2 mm in size to know the effects of ash on pore clogging. Thus, we rephrased the sentence as like [p.1] “Ash refers pyroclastic deposits with mean diameters smaller than 2 mm [9] and is produced from the combustion of surface fuels and organic matter in litter and duff layers.”
7. A suggestion to remove unnecessary/similar references and include only which have been actually used for this work. Current reference number (93) is very high for a journal article.
[Response] Thanks for pointing that out. We rearranged the references and consequently cited 79 papers.

Reviewer 2 Report
“Given that ash particles could not be distinguished from the soil particles based on color, the extent of ash clogging was indirectly estimated based on differences in the per-centage of stained (blue) pixels between the no rainfall and rainfall simulation experi-ments.”
This is the biggest problem in this study.
Other comment attached.

Author Response
Comments from Reviewer #2
1. [Title] Does soil pore clogging influence the Post-Fire Soil Hydrology? As seen in the abstract, it does not mean that. The title should be modified.
[Response] Thanks for your comment. We agreed on your opinion. Thus, we revised the title as “Exploring the Role of Ash on Pore Clogging and Hydraulic Properties of Ash Covered Soils under Laboratory Experiments”
2. [Abstract] Decrease? Need to be more specific in sentence “The rainfall simulations on ash-covered soils led to approximately 45.2% to 94.6% of the hydraulic conductivity of the bare soils, contributing to the combined effects of surface crust formation and soil pore clogging.”
[Response] Thanks for pointing that out. We rephrased the sentence in order to clarify the meaning as follow: “The rainfall simulation experiments also showed that ash cover led to the reduction of soil hydraulic conductivity, owing to the combined effects of surface crust formation and soil pore clogging”. In addition, we also rephrased the some part of abstract session.
3. [p. 2] As presented above, there are many related researches. “Soil pore clogging in fire-affected soils has received relatively far less attention than the presence of soil water repellency”
[Response] Thanks. We agreed on your comment. We’d like to point out that the studies on post-fire effects in soil science focused mostly on soil water repellency, thus, less attention was paid to pore clogging, compared to soil water repellency. We changed “far less” to “relatively less” in the sentence.
4. [p. 2] Compared with the above related studies, what problems remain unsolved? I think there has been a lot of attention. What are the problems in this research? Where is the innovation of this paper compared with other studies? “Laboratory experiments have been widely adopted to elucidate their isolated impact on post-fire hydrology by controlling as many factors as possible [32–34].”
[Response] Thanks for your comments. We rephrased the sentences by indicating the questions of this study. “Laboratory experiments have been widely adopted to overcome these limitations by controlling as many factors as possible [25,26]. They quantified the effects of ash on soil hydraulic properties, overland flow, and soil erosion. However, still few studies address the presence of pore clogging on ash covered soils and the effect of pore-blocking on soil hydraulic properties”
5. [p. 2] Who was affecting whom? Please rewrite. “Despite an increasing number of studies on the presence of soil hydrophobicity after a fire, the presence of soil pore clogging and its effects on post-fire hydrology have rarely been studied.”
[Response] Thanks. We rephrased the sentence based on your suggestion. “An understanding of how ash blocks the soil pore spaces, and subsequently, how pore clogging affects the soil hydraulic property is a vital subject in recent studies on post-fire hydrology”.
And, we rephrased the last paragraph in the introduction session. Please see the text.
6. [p. 3] How about raindrop size and uniformity coefficient? Can the relevant experimental requirements be met?
[response] Thanks for pointing that out. Based on your comment, we put the paragraph to explain the characteristics of raindrops as follow: “The characteristics of simulated rainfall are listed in Table 1. The average rainfall intensities were set at 21.87, 37.50, and 47.97 mm/h for 30 min, which are nearly comparable to the design storm in the Okgye region with a return period of 2, 10, and 50 years, respectively (hereafter R02, R10, R50) [41]. The raindrop size varied between 0.1 mm to 4.0 mm, which is apparent for most moderate rainfall cases in Korea [42]. The median values of simulated raindrops were 0.8, 1.2 and 1.9 mm for R02, R10, and R50, respectively. The corresponding kinetic energies of rainfall were 35.19, 83.55, and 134.08 J/m2/h, respectively. The values of the Cristiansen uniformity coefficient were 0.91 for R02, 0.95 for R10, and 0.98 for R50, which were in the acceptable range of published uncertainty [43,44].”
7. [p. 3] Remove what particles?
[Response] Thanks. We removed the parts of leaves and branches in the samples. Thus, we revised as like “woody particles”
8. [p. 4] please add reference basis
[Response] Thanks. We revised it, by referencing (Table 2)
9. [p. 4] How much is it? It should be written here. In “three rainfall intensities (2-, 10-, and 50-year return period)”
[Response] Thanks. We rearranged the [2.1] session, therefore, this was embedded in Table 1. We referred here the Table 1.
10. [p. 5] [3.1 session] This form of analysis is not necessary because it is already listed in the table 1.
[Response] Thanks for your comment. Based on your suggestion, we deleted this paragraph and then moved this part and [Table 1] to [2.1.1] session with brief explanation.
11. [p. 5] It does not need to be analyzed as a result. It may be better to introduce it as a test material. Moreover, too many references are cited in the results.
[response] Thanks. We moved this part into [Materials and Methods] session. And, we deleted the reference if not necessary.
12. [p. 5] Does not correspond to the following. What is the relationship between Rainfall characteristics and your title?
[Response] Thanks. In this study, we would like to examine the effects of rainfall intensities on pore clogging formation, and then compared the severity of pore clogging according to rainfall intensity. And, we moved this session into [Materials and Methods] session, because it is not result, but the experiment condition.
13. [p. 7] This is the biggest problem in this study. I don't agree with this method. “Given that ash particles could not be distinguished from the soil particles based on color, the extent of ash clogging was indirectly estimated based on differences in the percentage of stained (blue) pixels between the no rainfall and rainfall simulation experiments.”
[Response] Thanks for your comments. We fully agreed with your criticism. In recent years, high resolution images taken from SEM or CT techniques were widely used for directly quantifying pore geometry, pore connection, and soil microstructure. It seems to be a reliable and accurate method in many areas, particularly soil science. Despite many advantages in using SEM/CT-derived images, it has also limitations in use. The high cost of operations, maintenance, and equipment can restrict to practical use in research issues particularly with a wide variation of property. Therefore, we tried to employ the digital camera images for indirect measurement of soil pore space, instead of the high resolution images. The development of ICT-based technology, such as digital camera, can help to acquire a relatively high quality images with low cost.
Of cause, there were many limitations to use the indirect methods used in this study.
Thus, we pointed out 1) the reason to employ the appearance comparison technique in estimation of pore clogging, and 2) the limitations in use, and rephrased/put the sentences in the discussion session [4.1] , as like
“ Resin impregnation is more expensive and labor intensive than some traditional methods of measuring soil structure but offers the advantage of a more detailed assessment of soil pore distribution when incorporating image analysis [60]. In this study, the soil pore space was graphically identified using binary gradients between the stained and unstained pixels.
The appropriate way to quantify the soil pore clogging is to directly identify ash particles that stuck in pore space using the scanning electron microscope and X-ray computed tomography techniques. Due to high cost for operation and a wide variation of pore geometry in soil samples, it is not always possible to practically measure the presence and magnitude of pore blocking. Instead, indirect measurement with image analysis is often employed for the appearance analysis of pore space.
This paper proposed the reliable procedure to extract the pore space from digital camera images, assuming that the extent of soil pore is roughly the same for all soil section specimens. However, there were several limitations for the quantification of pore clogging. Ash particle sometimes cannot distinguished from soil solid in resin impregnated image. It led to but small errors in the estimation of pore space because the epoxy resin permeated near the edge of soil particles. In addition, it should be noted that the inter- and intra-particle pore spaces has not been examined, which contributes soil hydraulic properties. A further study is needed to improve the method of soil digital image acquisition and color information extraction.
“
Thanks.

Round 2
Reviewer 1 Report
Authors have responded to all my comment. I recommend the manuscript for publication.
Author Response
Comments from Reviewer #1:
Based on your comments, we carefully check the cited references and grammer errors.
Thank you for reviewing our manuscript
Reviewer 2 Report
The authors have addressed correctly many of the reviewers' comments. This is a much better manuscript than before. However, the research method needs further explanation. Once that is improved, I would recommend the journal accept the manuscript.
Specific comments below:
"the extent of ash clogging was indirectly estimated based on differences in the percentage of stained (blue) pixels between the no rainfall and rainfall simulation experiments"
The soil pore clogging by ash was indirectly estimated based on differences in the percentage of stained (blue) pixels between the no rainfall and rainfall simulation experiments. As we know, rainfall can also lead to soil pore clogging. Therefore, how to distinguish whether soil pore clogging is caused by rainfall or ash?
Author Response
Comments from Reviewer #2:
The authors have addressed correctly many of the reviewers' comments. This is a much better manuscript than before. However, the research method needs further explanation. Once that is improved, I would recommend the journal accept the manuscript.
Specific comments below:
"the extent of ash clogging was indirectly estimated based on differences in the percentage of stained (blue) pixels between the no rainfall and rainfall simulation experiments"
The soil pore clogging by ash was indirectly estimated based on differences in the percentage of stained (blue) pixels between the no rainfall and rainfall simulation experiments. As we know, rainfall can also lead to soil pore clogging. Therefore, how to distinguish whether soil pore clogging is caused by rainfall or ash?
[Response] Thanks for your comments. We agreed that the sentences mentioned was not clear, thereby it is difficult to identify the effects of ash on pore clogging, just based on the sentence.
Therefore, we rephrased in order to clearly describe the methodology used in pore clogging estimation, as follow:
[p. 7] “Pore plugging by ash particles was visualized with the aid of image analysis (Fig-ure 3). Given that ash particles could not be distinguished from the soil particles based on color, the extent of ash clogging was indirectly estimated based on differences in the percentage of stained (blue) pixels between the experiment. The volume densities for the bare soils (no ash treatment) was considered as the reference of soil macropores for each rainfall simulation.”
(…..)
“The differences in the volume density of the soil pores between the no rainfall and rainfall experiments demonstrated the effects of raindrop impact on migration of fine particles through the soil macropores……..”
(….)
“By comparing the extent of soil pores between the bare soil and ash treatments (Table 3), additional reduction in pore volume because of ash clogging was observed. …..”